# Discovery and Preliminary Characterization of Translational Modulators that Impair the Binding of eIF6 to 60S Ribosomal Subunits

**DOI:** 10.3390/cells9010172

**Published:** 2020-01-10

**Authors:** Elisa Pesce, Annarita Miluzio, Lorenzo Turcano, Claudia Minici, Delia Cirino, Piera Calamita, Nicola Manfrini, Stefania Oliveto, Sara Ricciardi, Renata Grifantini, Massimo Degano, Alberto Bresciani, Stefano Biffo

**Affiliations:** 1National Institute of Molecular Genetics, “Fondazione Romeo ed Enrica Invernizzi”, INGM, Via Francesco Sforza 35, 20122 Milan, Italy; pesce@ingm.org (E.P.); miluzio@ingm.org (A.M.); cirino@ingm.org (D.C.); calamita@ingm.org (P.C.); manfrini@ingm.org (N.M.); oliveto@ingm.org (S.O.); ricciardi@ingm.org (S.R.); grifantini@ingm.org (R.G.); 2Department of Translational and Discovery Research, IRBM S.p.A., Via Pontina km 30, 600, 00071 Pomezia (Roma), Italy; l.turcano@irbm.com; 3Biocrystallography Unit, Dept. of Immunology, Transplantation and Infectious Diseases, IRCCS Scientific Institute San Raffaele, Via Olgettina 58, 20132 Milan, Italy; minici.claudia@hsr.it (C.M.); degano.massimo@hsr.it (M.D.); 4DBS, University of Milan, Via Celoria 26, 20133 Milan, Italy

**Keywords:** iRIA, initiation, polysomes, eIF4E, RACK1, Shwachman–Diamond syndrome, eIFsixty-i

## Abstract

Eukaryotic initiation factor 6 (eIF6) is necessary for the nucleolar biogenesis of 60S ribosomes. However, most of eIF6 resides in the cytoplasm, where it acts as an initiation factor. eIF6 is necessary for maximal protein synthesis downstream of growth factor stimulation. eIF6 is an antiassociation factor that binds 60S subunits, in turn preventing premature 40S joining and thus the formation of inactive 80S subunits. It is widely thought that eIF6 antiassociation activity is critical for its function. Here, we exploited and improved our assay for eIF6 binding to ribosomes (iRIA) in order to screen for modulators of eIF6 binding to the 60S. Three compounds, eIFsixty-1 (clofazimine), eIFsixty-4, and eIFsixty-6 were identified and characterized. All three inhibit the binding of eIF6 to the 60S in the micromolar range. eIFsixty-4 robustly inhibits cell growth, whereas eIFsixty-1 and eIFsixty-6 might have dose- and cell-specific effects. Puromycin labeling shows that eIF6ixty-4 is a strong global translational inhibitor, whereas the other two are mild modulators. Polysome profiling and RT-qPCR show that all three inhibitors reduce the specific translation of well-known eIF6 targets. In contrast, none of them affect the nucleolar localization of eIF6. These data provide proof of principle that the generation of eIF6 translational modulators is feasible.

## 1. Introduction

Translational control is the process by which mRNAs are differentially decoded into proteins. Translation is a relatively slow and energetically demanding process. For this reason, the rate of translation adapts to extracellular conditions through a complex series of signaling pathways. Translation is divided in four phases: initiation, elongation, termination, and recycling. For any given mRNA, initiation is the rate-limiting process [1,2,3,4]. Growth factors and nutrients stimulate initiation by converging signaling cascades on eukaryotic initiation factors (eIFs). One of the best known pathways activated by insulin and growth factors is the PI3K-mTORC1 (mTOR complex 1) signaling network, which stimulates eIF4F formation. mTORC1 phosphorylates 4E-BPs, (eIF4E binding proteins), which release the cap complex binding protein eIF4E. Free eIF4E assembles in the eIF4F complex, which contains mRNA, the eIF4A helicase, and eIF4G. The eIF4F complex binds 43S ribosomal subunits, leading to the formation of 48S pre-initiation complexes and the subsequent activation of cap-dependent translation. eIF4F controls the translational efficiency of specific mRNAs downstream of mTORC1 activity, resulting in the induction of cell growth and cell cycle progression [5]. A parallel cascade that converges on translation is represented by the RAS/MAPK pathway. RAS activates the MAPK of Mnk1/2 kinases, which phosphorylate eIF4E [6]. eIF4E phosphorylation causes increased tumorigenesis through an unknown molecular mechanism [7]. Both pathways have attracted the attention of cancer biologists. As translation dysregulation is a widespread characteristic of tumor cells, therapeutic agents that target the initiation of translation can potentially function as anticancer drugs that are capable of overcoming intra-tumor heterogeneity [8].

The inhibition of mTORC1-dependent translation by rapamycin and its analogues is beneficial in selective cancers characterized by mTORC1 activation [9,10]. However, patients with RAS mutations are insensitive to mTORC1 inhibition [11], suggesting that other initiation factors must act in an mTOR-independent fashion. Along this line, novel inhibitors targeting the Mnk pathway are under development [8,12,13]. 

Another promising target is represented by eIF6. eIF6 was originally identified for its ability to inhibit the association of 40S and 60S ribosomal subunits into 80S, in vitro [14]. A small pool of nuclear eIF6 is essential for ribosome biogenesis [15]. In vivo, eIF6 is essential for efficient translation. Evidence that eIF6 is involved in the regulation of translation comes from the characterization of eIF6 +/− mice. As a matter of fact, mice that have half the levels of eIF6 do not increase protein synthesis in response to insulin and growth factor stimulation [16]. Subsequent studies have shown that eIF6 is necessary for the efficient translation of mRNAs containing upstream open reading frames (uORFs) or G/C rich sequences in their 5′UTRs [17]. Overall, eIF6 acts as a global regulator of metabolism [18,19]. 

eIF6 activity is heavily affected in tumor cells and its modulation has a potential value in both cancer and genetically inherited diseases. A high expression of eIF6 correlates with human cancer malignancy and progression [20,21,22,23]. Studies in mice have unequivocally shown that eIF6 levels control cancer progression and mortality. The tumorigenic potential of eIF6 is striking in a mouse model of lymphomagenesis in vivo. In this setting, expression of the Myc oncogene under the control of the enhancer of IgH (Eµ-Myc) in the B-cell lineage drives a lethal lymphoma, similar to B-cell lymphomas, with a median survival of only 4 months. Eµ-Myc/ eIF6 +/− mice have increased survival, up to 1 year [24]. Overall, these data suggest that the modulation of the antiassociation activity of eIF6, which is known to be regulated by growth factor signaling pathways [25,26], can have a specific effect in tumor environments. 

In addition to cancer cells, eIF6 antiassociation activity is pivotal in the phenotype caused by loss of function mutations of SBDS and eFl1. SBDS is a ribosome-associated factor that mediates 60S biogenesis and its export from the nucleus to the cytoplasm [27]. In humans, mutations of the SBDS gene cause Schwachman–Diamond syndrome, which is an inherited disease with a multiorgan phenotype [28]. Importantly, eIF6 gain-of-function mutants rescue the phenotypes of both Elf1 [29] and SBDS deletions [27,30,31,32,33]. In conclusion, finding modulators of eIF6 activity can have a tremendous impact on specific pathological situations.

In this study, we developed a novel method for finding modulators of eIF6 binding to the 60S subunit [34] and performed an automated screening of small compounds. Here, we describe the screening and identification of three compounds that have a biological activity that can modulate the binding of eIF6 to the 60S. Consequently, all compounds were found to modulate translation. In addition, in spite of having different actions on cell growth and global translation, all three compounds reduced the translation of specific eIF6 targets. This study provides proof of concept for both the feasibility and value of modulating eIF6 binding to the 60S in vivo.

## 2. Materials and Methods

### 2.1. Screening Library and eIF6 Purification

The library tested in the present work is a subset of the CNCCS (The National Consortium and Collection of Chemical Compounds) collection (c. 150,000 compounds, www.cnccs.it). The tested subset was manually curated to include FDA/EMA (Food and Drug Administration/European Medicines Agency) approved drugs, compounds that successfully passed Phase I but have not, or have not yet, reached the market, compounds that were previously reported to be active against post-translational modifying enzymes, compounds that were predicted to bind RNA, and their analogs. The identified compounds were cherry-picked from 10 mM DMSO solutions and arrayed for testing in the present work. The preparation and purification of eIF6 and of the 60S subunit were performed as previously described [34].

### 2.2. 60S–eIF6 Homogeneous Binding Assay

The following materials were purchased from Sigma-Aldrich (St. Louis, MO, USA): MgCl_2_, BSA (Bovine Serum Albumin), and Tween-20. AlphaScreen streptavidin donor beads and the AlphaScreen Histidine (Nickel chelate, St. Louis, MO, USA) Detection kit were purchased from PerkinElmer (USA). The 60S subunit was biotinylated (60SBio) using the EZ-Link™ Sulfo-NHS-LC-Biotinylation Kit (ThermoFisher Scientific, Waltham, MA, USA) according to the manufacturer’s instruction. Compounds from 10 mM stock solutions were transferred to assay plates by acoustic transfer (EDC Biosystems, Milmont, CA, USA). The 60S–eIF6 binding assay was performed in AlphaPlate-384 plates (PerkinElmer, Shelton, CT, USA). The following components were added to the plates to a final volume of 10 μL and at the following concentrations: eIF6 75 nM, 60Sbio 0.5 nM in PBS (Phosphate-buffered saline) (pH 7.4), 5 mM MgCl_2_, 0.2% BSA, 0.02% Tween-20. After 60 min of incubation at room temperature, 5 µL/well of acceptor beads (20 µg/mL diluted in Alpha detection buffer) were added. After an additional 60 min of incubation at room temperature, 5 µL/well of donor beads (20 µg/mL diluted in Alpha detection buffer) were added. The reaction was left to incubate for 60 min at room temperature; then, the alpha-screen signal was acquired by an EnVision plate reader (PerkinElmer, Waltham, MA, USA). The results were analyzed using Prism (GraphPad, San Diego, CA, USA) and Vortex (Dotmatics, Bioshops Stortford, UK). Dose–response curves were fitted by four-parameter logistic regression.

### 2.3. 60S–eIF6 Counter-Screen Assay 

The Alpha beads and the 6His-streptavidin linker were provided within the AlphaScreen Histidine (Nickel chelate, St. Louis, MO, USA) Detection kit (PerkinElmer, USA). Compounds from 10 mM stock solutions were transferred to assay plates by acoustic transfer (EDC Biosystems, Milmont, CA, USA). The assay was performed in AlphaPlate-384 plates (PerkinElmer, USA). The following components were added to the plates to a final volume of 10 μL: 20 nM of 6His-streptavidin linker in PBS (pH 7.4), 5 mM MgCl_2_, 0.2% BSA, and 0.02% Tween-20. After 60 min of incubation at room temperature, 5 µL/well of acceptor beads (20 µg/mL diluted in Alpha detection buffer) were added. After an additional 60 min of incubation at room temperature, 5 µL/well of donor beads (20 µg/mL diluted in Alpha detection buffer) were added. The reaction was left to incubate for 60 min at room temperature; then, the alpha-screen signal was acquired by an EnVision plate reader (PerkinElmer, Waltham, MA, USA). The results were analyzed using Prism (GraphPad, San Diego, CA, USA) and Vortex (Dotmatics, Bioshops Stortford, UK). Dose–response curves were fitted by four-parameter logistic regression.

### 2.4. Chemistry

Compounds were obtained from commercial suppliers and were tested without further purification. Purity was determined using MS and UPLC. UPLC-MS analyses were performed on a Waters Acquity UPLCTM, equipped with a diode array and a ZQ mass spectrometer, using a Waters BEH C18 column (1.7 μm, 2.1 × 50 mm). The mobile phase comprised a linear gradient of binary mixtures of H_2_O containing 0.1% formic acid (A), and MeCN containing 0.1% formic acid (B). The following linear gradient was used (A): 90% (0.1 min), 90%–0% (2.6 min), 0% (0.3 min), 0%–90% (0.1 min). The flow rate was 0.5 mL/min. The purity of final compounds was in all cases ≥95%. 

### 2.5. iRIA

The general procedure was performed as previously described with some modifications [34]. Briefly, 96-well plates were coated with 0.5 nmol of purified ribosomes diluted in 50 µL of PBS 1× −0.01% Tween-20, O/N at 4 °C in a humid chamber. The coating solution was removed and unspecific sites were blocked with 10% BSA powder in PBS 1× −0.01% Tween-20 for 30 min at 37 °C. The plate was washed with 100 microliters/well with PBS-Tween. Biotynilated eIF6 was resuspended in a reaction mix to the final concentration of 75 nM with 5 mM MgCl_2_, 0.2% BSA, 2% DMSO, PBS-0.01% pH 7.4, and 0.02% Tween-20 to reach 50 µL of final volume/well. The reaction mix was added in the well and was incubated with ribosomes for 1 h at room temperature. To remove unbound proteins, each well was washed 3 times with PBS-Tween. HRP-conjugated streptavidine was diluted 1:5000 in PBS-Tween, and 50 µL were incubated in the well for 30 min at room temperature. Binding was revealed as previously described [34]. Compounds were added to the mixture during the incubation with eIF6 or as specified. 

### 2.6. Cells and Vitality Curves

The following human cell lines were used for vitality curves: Colo205, Ovcar-8, H266, and MDA-MB-468. All cell lines were obtained from the American Type Culture Collection (ATCC; www.atcc.org) and cultured in RPMI-1640 medium (Euroclone, Pero, Italy) supplemented with 10% fetal bovine serum (FBS) and penicillin/streptomycin/glutamine solution (GIBCO) at 37 °C and 5% CO_2_. Cells were routinely tested for mycoplasma contamination. Cell viability was performed with the PrestoBlue cell viability reagent (Invitrogen) according to manufacturer’s instructions. Briefly, 1000 cells were plated at 100 µL/well in a 96-well plate; then, they were assayed after 48 and 96 h.

### 2.7. Puromycin Labelling 

Cells were grown in their medium supplemented with 10% fetal bovine serum (FBS—#ECS0180L, Euroclone, Pero, Italy) and 5 mL of penicillin/streptomycin/glutamine solution (GIBCO, Waltham, MA, USA), until they were subconfluent. For protein synthesis measurement, the cells were treated at 65%–70% confluence with compounds (eIF6ixty-i) at the indicated concentration for 30 min. Next, puromycin (#A1113803, Thermofisher Scientific, Waltham, MA, USA) was added to the final concentration of 5 μg/mL and left for 30 min. At the end of the incubation period, cells were washed twice in cold PBS and then lysed for puromycin detection by Western blotting using anti-puromycin antibodies (Sigma-Aldrich, # MABE343). The quantitation of puromycin incorporation was performed using the Image J software (Fiji version (http://fiji.sc), Bethesda, MD, USA). At least three separate experiments for each compound were performed. Statistical analysis was performed by a two-sample t-test comparing the treatment to the DMSO-only control that was always run in the same gel with the compounds.

### 2.8. Polysomal Profiles

Polysomal profiles were performed as previously described [16]. In brief, cells were lysed in 30 mM Tris-HCl, pH 7.5, 100 mM NaCl, 30 mM MgCl_2_, 0.1% NP-40, 10 mg/mL cycloheximide, and 30 U/mL RNasin. After centrifugation at 12,000 rpm for 10 min at 4 °C, cytoplasmic extracts with equal amounts of RNA were loaded on a 15%–50% sucrose gradient and centrifuged at 4 °C in a SW41Ti Beckman rotor for 3 h 30 min at 39,000 rpm. Absorbance at 254 nm was recorded by BioLogic LP software (BioRad, Hercules, CA, USA), and 10 fractions (1.5 mL each) were collected for subsequent proteins [35] and RNA extraction.

### 2.9. Analysis of Translated mRNAs

mRNAs collected from fractions of sucrose gradient during polysome profiling analysis were divided into subpolysomal, (from the top of the gradient to the 80S peak), light polysomes, and heavy polysomes mRNAs. Samples were incubated with proteinase K and 1% SDS (Sodium dodecyl sulfate) for 1 h at 37 °C. RNA was extracted by the phenol/chloroform/isoamyl alcohol method. After treatment of RNA with RQ1 RNase- freeDNase (Promega, Madison, WI, USA), reverse transcription was performed according to SuperScript III First-Strand Synthesis kit instructions (Thermo Fisher Scientific, Waltham, MA, USA). Complementary cDNA (100 ng) was amplified with the appropriate primers using StepOne Plus Real-Time PCR System (Thermo Fisher Scientific,). Taqman probes specific for eIF6 (Hs00158272_m1), rpL5 (Hs03044958_g1), and 18S rRNA were used as internal standards. SYBR green specific probes were used for 18S rRNA (Fw 5′-CATGCAGAACCCACGACAGTAC-3′, Rw 5′-CCTCACGCAGCTTGTTGTCTA-3′), Cebpd (Fw 5′-ATCGACTTCAGCGCCTACAT-3′, Rw 5′-GCTTTGTGGTTGCTGTTGAA-3′), Cebpb (Fw 5′-CAAGCTGAGCGACGAGTACA-3′ Rw 5′-AGCTGCTCCACCTTCTTCTG-3′), ATF4 (Fw 5′-GAGCTTCCTGAACAGCGAAGTG-3′ Rw 5′-TGGCCACCTCCAGATAGTCATC-3′), Actin (FW 5′-GCACTCTATGCCAACACAGTGC-3′; RW 5′ CCTGCTTGCTGATCCACATCTG-3′), and GAPDH (Glyceraldehyde 3-phosphate dehydrogenase) (FW 5′ CATGACAACTTTGGCATTGTG-3′; RW 5′ GTTGAAGTCGCAGGAGACAAC-3′). A Taqman probe specific for rpS19 (Mm00452264_m1) was also used. The quantification cycle (Cq) values were acquired for each sample with the StepOne Software V2 3. (Thermo Fisher), with the following cycling conditions: 10 min at 95 °C for initial denaturation, and an amplification program of 40 cycles at 95 °C for 15 s and 60 °C for 1 min. The fluorescence threshold was manually set above the background level. Target mRNAs quantification was performed by using the ΔΔCt method with 18S rRNA as an internal standard. For the analysis of targets from polysome fractions, the data are quantitated as the percentage of expression in each fraction, normalized to the total amount of the target. 

### 2.10. Immunofluorescence 

Immunofluorescence was performed using an anti-eIF6 antiserum previously described [36]. All procedures were as previously described [15,37]. Nucleophosmin (NPM) was used as an independent nucleolar marker, and phalloidin was used for the detection of the cytoskeleton. Images were acquired using a filterless laser-scanning confocal microscope (Leica TCS SP5) using excitation wavelengths of 488 nm for green fluorescence, 561 nm for red fluorescence (immunofluorescence), and 405 nm for DAPI (4′,6-diamidino-2-phenylindole) (Sigma, St. Louis, MO, USA) nuclear labeling. All the images were further processed with Photoshop CS6 (Adobe, Berkeley, CA, USA) software.

## 3. Results

### 3.1. eIF6-60S Homogenous Binding Assay Optimization

The original assay for eIF6 binding to the 60S has been previously described [34]. In short, pure 60S ribosomal subunits were immobilized on microwell plates and reacted with full length recombinant eIF6 [34,38]. The screening of compound collections against one target requires the setup of homogeneous and automation suitable assays. To this purpose, we envisaged the use of the AlphaScreen technology [39] as a homogeneous proximity detection technique. To obtain the desired 60S–eIF6 binding signal, 60S subunits were biotinylated (60S-bio) using a chemical biotinylation kit (see methods) in order to be captured by streptavidin conjugated AlphaScreen donor beads (SA-Don) (Figure 1A), while His-tagged eIF6 was stained by Ni-NTA conjugated AlphaScreen acceptor beads (His-Acc) (Figure 1A). Within this assay setup, the AlphaScreen proximity signal is detectable only when eIF6 is bound to the 60S-Bio (Figure 1A). The binding reaction occurs in solution inside each well of the 384 well plate. We anticipate that the concentration of SA-Don beads is a limiting factor in the present assay setup. Indeed, while only one His-tag is present for each eIF6 molecule, making its saturation easy by His-Acc beads, several biotins are present for each 60S-Bio, making its saturation through SA-Don beads more complicated. In an attempt to optimize the assay set up (S-1), we set out to cross SA-Don and His-Acc beads and verified the linear signal-to-noise relationship. With the aim of making the present approach cost-effective, we arbitrarily set the concentration of SA-Don beads and of His-Acc beads at 20 µg/mL. In order to optimize the concentration of 60S-Bio and eIF6 to be used in the assay, we carried out a titration of 60S-Bio against a fixed concentration of 25 nM eIF6 (Figure 1B). The results showed a hook effect at approximately 0.8 nM 60S-Bio, suggesting that this is the concentration at which SA-Don beads become limiting. Having fixed the 60S-Bio concentration to 0.5 nM, the Kd of eIF6 for 60S was determined by incubating a serial dilution of eIF6 with 0.5 nM 60S-Bio, resulting in a calculated Kd of 40.0 ± 4.2 nM (Figure 1C). Although the K_d_ value is commonly used as the assay concentration, we decided to use a final eIF6 concentration of 75 nM to improve the signal/background ratio. Finally, to prove that the disruption of the 60S-Bio/eIF6 complex was detected by the assay, a serial dilution of non-biotinylated 60S was run against the assay as optimized. The obtained results were normalized between 100% displacement (no eIF6) and 0% displacement (the binding reaction). The resulting curve demonstrated that the assay is suitable for detecting disruptors of 60S–eIF6 binding (Figure 1D). The IC50 for non-biotinylated 60S was calculated to be 0.28 ± 0.06 nM, suggesting that in these assay conditions, we are titrating the concentration of 60-Bio.

### 3.2. Hit Identification 

A collection of 2977 bioactive compounds present in our library was screened at 10 µM using the protocol described above. The collection includes FDA/EMA-approved drugs, compounds that successfully passed Phase I but did not, or have not yet, reached the market, compounds previously reported to be active against post-translational modifying enzymes, and compounds that were predicted to bind RNA. The Z’ values [40] were found to be greater than 0.5 for all screening plates, indicating that the assay was sufficiently robust to be used for hit identification. The distribution of the compound activities converges to normal distribution (or Gaussian distribution) (Figure 2A). The raw data and the name of the compounds that were not selected for further studies cannot be disclosed due to IP policies. Compounds with an activity equal to or greater than 50% were considered hit compounds. After having discarded false positives by retesting in triplicate, seven compounds, corresponding to 0.23% of the total, were identified as active in the primary screening. Their purity (>95%) and identity were verified by LC-MS. To confirm that they were actual hits, the seven compounds were tested in a dose–response fashion, resulting in the confirmation of three chemotypes in the primary assay with IC50 values ranging from 1 to 10 µM (Figure 2B–D) and not interfering with the AlphaScreen technology. The interference screening assay was carried out by measuring the interference of the compounds when the SA-Don and His-Acc beads are brought together by a 6His-streptavidin linker. Structural analysis revealed the presence of a compound previously approved for the treatment of leprosy (clofazimine, (E)-N,5-bis(4-chlorophenyl)-3-(isopropylimino)-3,5-dihydrophenazin-2-amine, eIFsixty-1) with an IC50 of 0.98 ± 0.07 µM (Figure 2B), a compound (14-benzoylacenaphtho[1,2-d]benzo[4,5]thiazolo[3,2-a]pyridin-13-ium, eIFsixty-4) with an IC50 of 5.38 ± 0.1 µM (Figure 2C) that was previously reported to be active against PRMTs (Protein arginine methyltransferases) [41], but whose structure was judged not to be lead-like and suitable only for in vitro investigations, and a compound with an IC50 of 1.05 ± 0.16 µM (Figure 2D) pertaining to the class of predicted RNA binders (5-(4-benzylpiperazin-1-yl)-2-(2-phenylcyclopropyl)oxazole-4-carbonitrile, eIFsixty-6).

### 3.3. Independent ELISA Validation

The screening was based on the proximity principle of the α-screening technology. However, the interaction between eIF6 and the 60S can be also hampered by the large size of acceptor and donor beads. Thus, ELISA assays were performed to independently score for the capability of “eIFsixty-i” compounds to inhibit the binding of eIF6 to 60S ribosomal subunits. We employed the previously developed iRIA technique [34]. Briefly, pure ribosomes were used to coat microwell plates and were preincubated with hit compounds at the IC50 concentration for up to 5 min (Figure 3A). Next, the full-length eIF6 protein was added to a concentration of 75 nM, and binding was measured in triplicate (Figure 3B). Controls were either the addition of no compound or the addition of a compound isolated from the primary screening, but not confirmed (eIFsixty-neg). This assay confirmed the capability of eIFsixty-i compounds to strongly and specifically inhibit eIF6 binding to the 60S, reaching binding inhibitions ranging from 75% for eIFsixty-6 to 90% for eIFsixty-4 (Figure 3C).

### 3.4. Viability Assays

Next, we wanted to understand if eIFsixty-i compounds were able to affect cell growth. We randomly selected four cell lines, Colo205, Ovcar-8, MDA-MB468, and H226, and measured the effects of the eIFsixty-i compounds at 48 and 96 h after administration at two concentrations, i.e., the IC50 as calculated by ELISA assays and the 0.2 × IC50. These concentrations were arbitrarily chosen to define potential toxicity effects. According to the data presented in Figure 4, eIFsixty-4 is a strong cytostatic factor for all the cell lines tested, acting at both IC50 and 0.2 × IC50 concentrations. eIFsixty-1 acts as a modest cytostatic, with effects mostly visible at 96 h, and not in all the cell lines analyzed. eIFsixty-6 showed either no effects or a mild increase of viability at 48 h, followed by either some toxicity (H226 cell line) or no effects. Taken together, these data suggest that eIFsixty-4 may be a strong inhibitor of translation, whereas for the other two compounds, the effects may be more dependent on the cell line, on the concentration, and on the time of delivery.

### 3.5. Puromycin Assays Show that eIFsixty-4 is a Strong Inhibitor of Global Translation, Whereas eIFsixty1 and eIFsixty-6 May Act As Mild Modulators

Subsequently, puromycin assays were performed in order to establish whether the compounds were cell permeable and/or able to inhibit global translation. For this and the following experiments, we focused on the H226 cell line, for which all of the compounds had an effect at 96 h. Cells were cultured at 75% confluency and pretreated for 10 min with either eIFsixty-1, eIFsixty-4, or eIFsixty-6 at the IC50 concentration. A representative triplicate is presented in Figure 5. Briefly, eIFsixty-4 strongly inhibited the incorporation of puromycin, eIFsixty-1 showed a trend of inhibition of puromycin incorporation, and eIFsixty-6 had the opposite effect. Quantitation of the experiment is shown in panel 5B. eIFsixty-4 behaves as a translational repressor. Concerning eIFsixty-1 and eIFsixty-6, we suggest that they may affect either the specific translation or global translation, depending on the cell line context.

### 3.6. Polysomal Assays Show that eIFsixty-4 Inhibits Initiation of Translation

We used polysomal profiling to assess the effect of the hit compounds. In line with the puromycin incorporation experiments, cells were cultured at 75% confluency and pretreated for 10 min with eIFsixty-1, eIFsixty-4, or eIFsixty-6 at the IC50 concentration. Cytosolic extracts were prepared and loaded on 17%–50% sucrose gradients. The results are shown in Figure 6. In line with all the previous results, the administration of eIFsixty-4 caused a drastic increase in the 80S peak and a reduction in the polysomal area, thus indicating that the inhibition of translation occurs at the initiation stage. Again, the effects of eIFsixty-1 and eIFsixty-6 were markedly different: little or no changes in the 80S peak were observed, suggesting that the two compounds may act on the translation of specific mRNAs.

### 3.7. All Compounds Modulate the Polysomal Association of Known eIF6 Target mRNAs

eIF6 modulates the translation of mRNAs containing uORFs and 5′UTR G/C-rich regions, including lipogenic factors CEBPβ, CEBPδ, and the transcription factor ATF4 [17]. We purified heavy polysome, light polysome, and subpolysome fractions, which should contain highly translated, averagely translated, and non-translated mRNAs, respectively, from polysomal profiles, such as those described in Figure 6. Then, we analyzed the abundance of several mRNAs: eIF6 itself and rpL5, which are not regulated by eIF6 activity and ATF4, CEBPβ, and CEBPδ, which are eIF6 targets [17]. The data are displayed in Figure 7. In short, all compounds, although to a different extent, induce a strong shift from the polysomal peak to the subpolysomal peak of all eIF6 target mRNAs, i.e., ATF4, CEBPβ, and CEBPδ. The effect on the polysomal association of mRNAs that are not eIF6 targets was more variable and, in general, much lower. eIFsixty-1 had basically no effect. eIFsixty-4 inhibited also the translation of eIF6 mRNA, which is in line with the observation that it acts as a strong translational inhibitor; however, it did not affect the translation of rpL5 mRNA. eIFsixty-6 modified the localization of eIF6 and rpL5 mRNAs, leading to a slight increase in the association with polysomes of rpL5, but not of eIF6. Taken together, these data are in line with all previous experiments and can be interpreted as follows: all compounds affect eIF6 targets. eIFsixty-1 and eIF-sixty6 may spare the translation of non-eIF6 targets.

### 3.8. Nucleolar Structure is not Affected by eIFsixty-i Compounds

eIF6 is necessary for 60S ribosome biogenesis [15]. To have an initial clue of the possibility that eIFsixty-i compounds could alter rRNA maturation by causing ribosomal stress, we stained for nucleolar eIF6. None of the hit compounds altered the nucleolar structure (Figure 8), suggesting that they do not affect ribosomal biogenesis. Future studies are required to further confirm this hypothesis.

## 4. Discussion

We provided proof-of-principle that it is feasible to target the activity of eIF6 and to affect translation. Small molecules targeting the translation machinery can be useful in the treatment of a variety of human diseases including cancer, viral infection, monogenic diseases, and others. For a long time, targeting of the translational machinery has been considered a “no-go” strategy because it was assumed that, given the essential and ubiquitous role of translation, the toxic side effects would have been predominant. In reality, in the last 20 years, a number of studies have overcome this rather commonplace and wrong belief. Long ago, it was reported that mRNA and protein levels are not matched [42]. After that, an overwhelming series of reports has identified regulatory sequences in the mRNA that interplay with initiation factors and are responsible for the tight regulation of translation downstream of specific signaling pathways [8,43]. Recent examples of the fact that biologically relevant gene expression is mainly regulated at the translation level include the response of immune cells to T-cell receptor stimulation [44], immunosuppression [45], neuronal activity [46], and many others [47]. In conclusion, targeting the activity of the translational apparatus results in much more specific effects than what was expected.

Another long-held belief was that the direct targeting of initiation factors was not feasible and was rather achieved by inhibiting signaling pathways converging on them. The most spectacular pathway inhibitors acting on translation are represented by mTORC1 inhibitors, eIF2 kinase inhibitors, and, more recently, Mnk inhibitors [8]. However, eukaryotic initiation factors perform mechanistic steps downstream of signaling cascade that include the interaction with specific RNA sequences or protein partners. More recently, exploiting this molecular feature, effective pharmacological compounds directly targeting the translational machinery have been developed. Among them, we include ISRIB (integrated stress response inhibitor) that binds eIF2B [48,49] and amino-rocaglates that directly inhibit eIF4A [50]. Several other strategies for other initiation factors have been attempted or are in progress [8].

Eukaryotic initiation factor 6 is a powerful ribosome-associated factor that performs two functions: it acts as an essential trans-actor factor in the generation and quality control of 60S subunits [15,51] and as a translation factor in the cytoplasm. eIF6 is essential for efficient growth factor-stimulated translation [16,17,24,25,26,52,53,54]. It is assumed that the biological activity of eIF6 resides in its capability to bind in a regulated fashion 60S ribosomal subunits, as it was shown originally by biochemical studies [14] and later confirmed by structural work [55]. Indeed, several studies have demonstrated that the release of eIF6 from the 60S is fundamental for ribosome maturation [31,51]. Importantly, the lack of eIF6 release from 60S subunits is part of the pathogenic mechanism of Swachman–Diamond syndrome [27,31,33], thus making this process a very specific target for intervention. Concerns of side effects due to targeting eIF6 are mitigated by the fact that only 20% of the protein is essential for survival [16] and by the fact that heterozygous mice, which have 50% of eIF6, are healthier than wild-type animals in several contexts [24,52]. In short, a partial targeting of the antiassociation activity of eIF6 is beneficial for at least two pathological contexts: Schwachman–Diamond syndrome and cancer cells. In this work, we have provided evidence that this targeting is feasible and paved the way for further studies.

This study represents an important starting point for future studies addressing the possibility to pharmacologically inhibit the activity of eIF6. Some considerations should be made at this point. To begin with, it would be interesting to extend the number of compounds that can be screened. In this specific context, the assay that we have developed, *i*RIA (in vitro Ribosomes Interaction Assay) [34], is highly reliable and flexible. Firstly, this is because to our knowledge, it is the only assay that measures with high accuracy eIF6 binding to 60S ribosomes. Secondly, it can be easily modified in order to adapt for other factors that may modulate the binding of initiation factors to ribosomes. Lastly, it can be used for the precise quantitation of free 60S ribosomal subunits as done in the past [38].

Possibly, a partial modulator of the binding of eIF6 to 60S can be found and may have more specific and interesting effects on translation. In particular, our work shows that some targets are not affected by eIFsixty-i compounds. An interesting example is represented by rpL5; the lack of an effect of eIFsixty-1 and eIFsixty-4 on rpL5 translation may be because this mRNAs is a TOP mRNA, very abundant, and under the control of mTOR signaling [4], whereas, as far as we know, eIF6 activity is totally independent of mTOR signaling [52]. Some modulators may increase the binding of eIF6 to 60S. Presently, compounds showing a negative regulation are within the normal Gaussian distribution, suggesting that negative values are accountable to variability, but more studies must be performed.

Currently, we do not know whether the identified compounds bind 60S subunits or eIF6. Of course, this fact may explain the differences found on their effect on translation and viability. This important aspect must be addressed in future studies. It should be also added that effects on translation may be secondary to effects on cell cycle. However, given the fact that the compounds have been isolated for their ability to block eIF6–60S interaction, and that experiments on translation have been performed upon short-term treatment, i.e., one hour, this possibility is not likely. Preliminary data have shown that short-term treatment of eIFsixty compounds does not displace eIF6 from the nucleolus. Obviously, this finding does not rule out the possibility that these compounds are affecting also ribosome biogenesis if applied for longer periods, or that they might interfere also with rRNA maturation. However, it has to be noted that the amount of eIF6 required for nucleolar ribosome maturation was estimated to be not more than 25% of the total pool [16]. This said, the effects on ribosome biogenesis must be studied in more detail. We conclude that this is the first report that shows a successful pipeline for the identification of eIF6 modulators. It is an important starting point for future studies.

## Figures and Tables

**Figure 1 cells-09-00172-f001:**
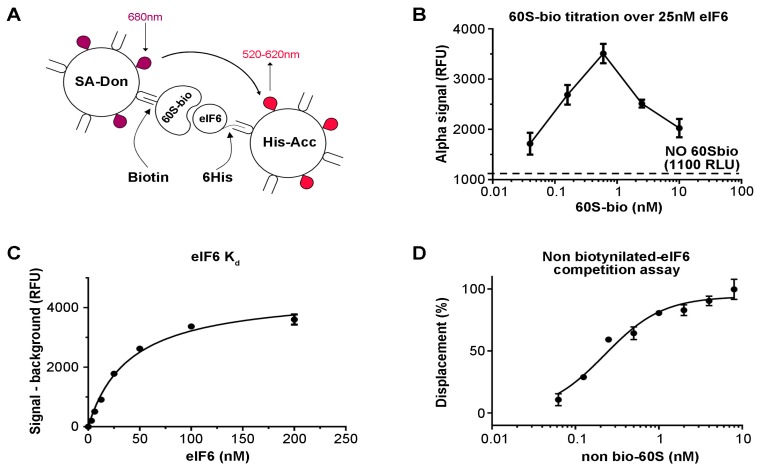
Optimization of the eukaryotic initiation factor 6 (eIF6)–60S binding assay. (**A**) Schematic drawing of AlphaScreen technology. (**B**) Optimization of the 60S-bio signal against a fixed concentration of eIF6. (**C**) eIF6 K_d_ value measurement. (**D**) Titration of non-biotinylated 60S displacement of eIF6 from biotynilated 60S. The results show that the assay is suitable for detecting disruptors of the 60S–eIF6 binding.

**Figure 2 cells-09-00172-f002:**
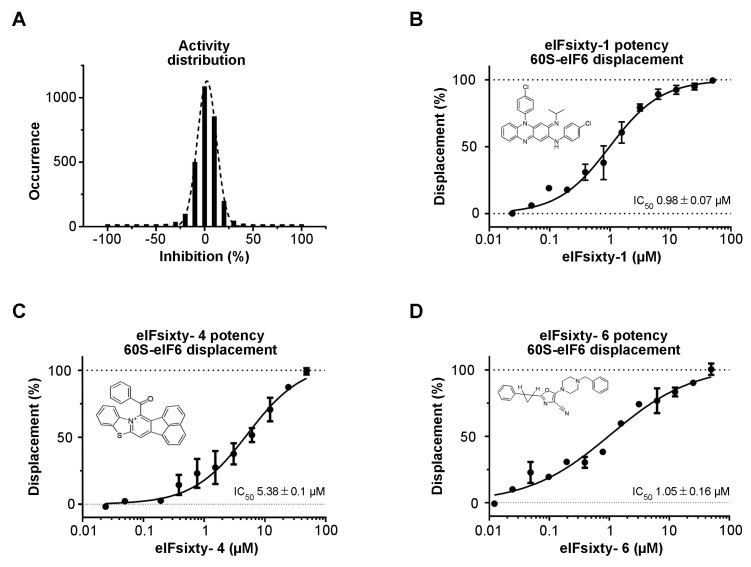
Hits screening. (**A**) Distribution of the activities of eIFsixty-i compounds. (**B**–**D**) Dose–response curve showing the eIF6–60S complex disruption potency of the selected eIFsixty-i compounds.

**Figure 3 cells-09-00172-f003:**
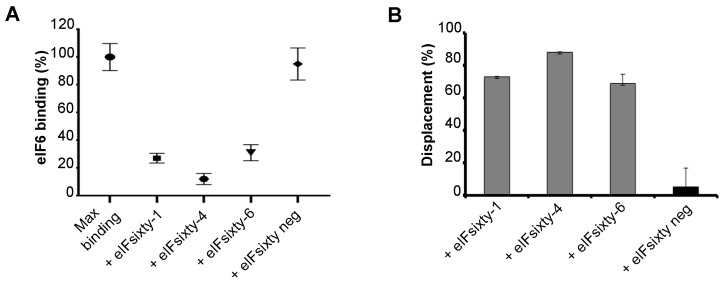
eIF6–60S binding evaluation in response to eIFsixty-i compounds by ELISA assay. (**A**) Relative eIF6–60S binding upon eIFsixty-i compounds (eIFsixty-1, eIFsixty-4, and eIFsixty-6) addition. eIFsixty-neg is considered as negative control. (**B**) Displacement of eIF6 from the 60S by eIFsixty-i compounds (eIFsixty-1, eIFsixty-4, and eIFsixty-6).

**Figure 4 cells-09-00172-f004:**
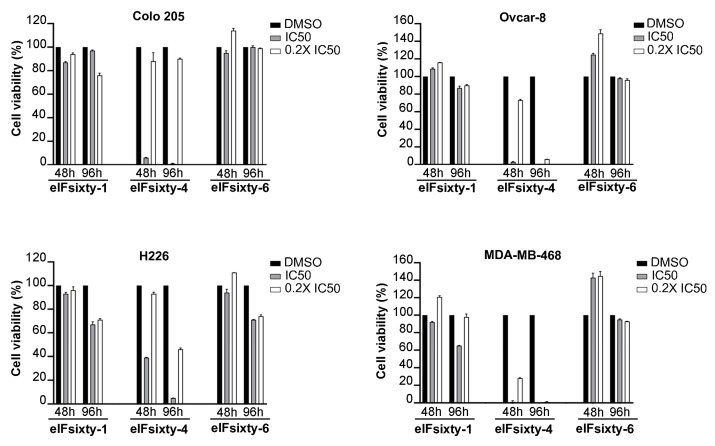
Cytostatic effects of eIFsixty-i compounds in cancer cell lines. Cell viability of Colo205, Ovcar-8, H266, and MDA-MB468 cells after administration of eIFsixty-i compounds (eIFsixty-1, eIFsixty-4 and eIFsixty-6). Data are presented as means ± SD.

**Figure 5 cells-09-00172-f005:**
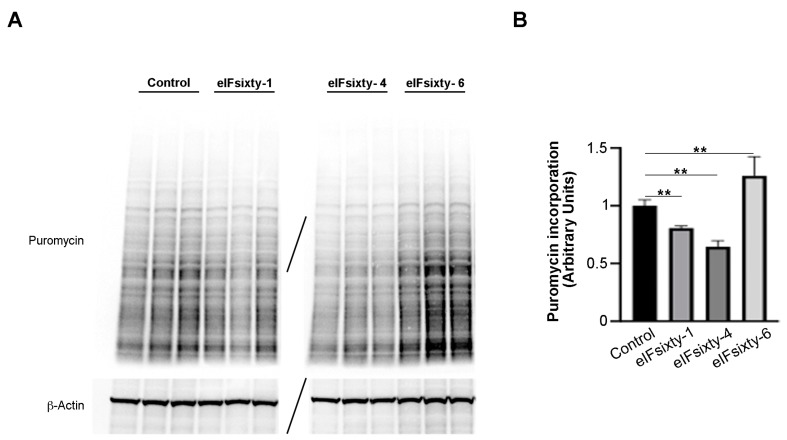
Protein synthesis significantly decreases upon eIFsixty-4 treatment. (**A**) Representative immunoblots of three independent experiments showing puromycin incorporation in H226 cells in the absence (control) or presence of eIFsixty-i compounds (eIFsixty-1, eIFsixty-4, and eIFsixty-6). (**B**) Quantitation of puromycin/actin. For quantitation, the puromycin labeling was measured by Image J. Three separate points of a triplicate were averaged and compared to the control triplicate, DMSO only, run in the same gel. T-test student, equal variance was applied. ** *p*-values, ≤0.01.

**Figure 6 cells-09-00172-f006:**
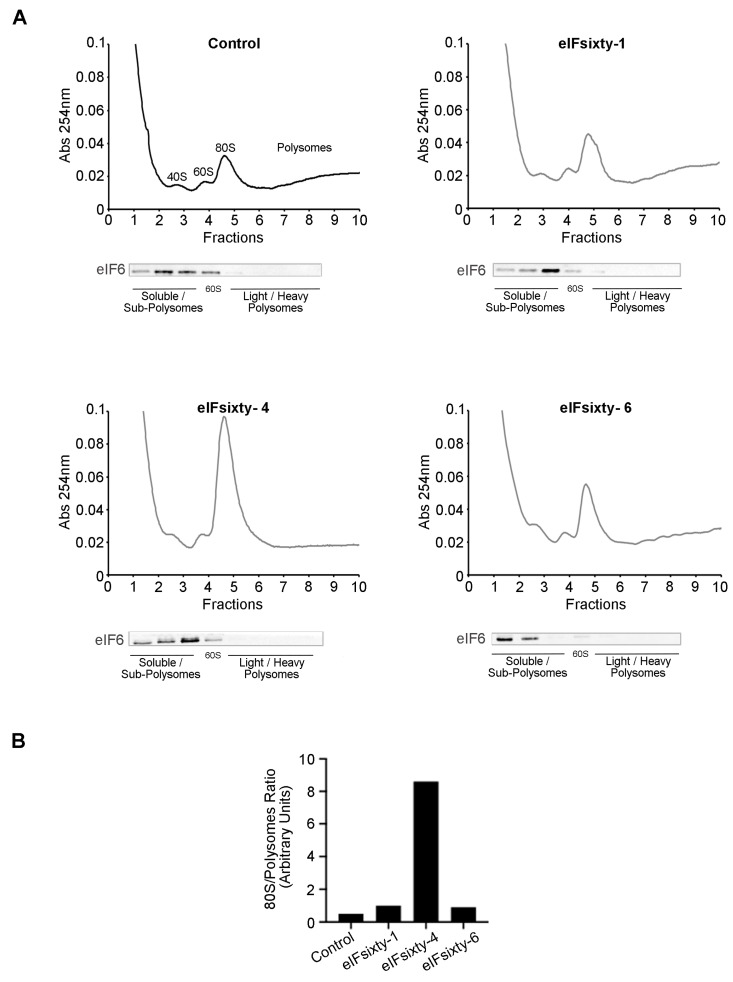
The eIFsixty-4 compound inhibits translation at the initiation stage. (**A**) Representative polysomal profiles of three independent experiments of H266 cells in the absence (control) or presence of eIFsixty-i compounds (eIFsixty-1, eIFsixty-4, and eIFsixty-6). The 80S peak is a marker of inactive translation. Upon eIFsixty-4 administration, the 80S peak dramatically increases and polysomes drop. Each polysomal fraction was assessed for eIF6 protein abundance. (**B**) Quantitation of 80S/polysomes.

**Figure 7 cells-09-00172-f007:**
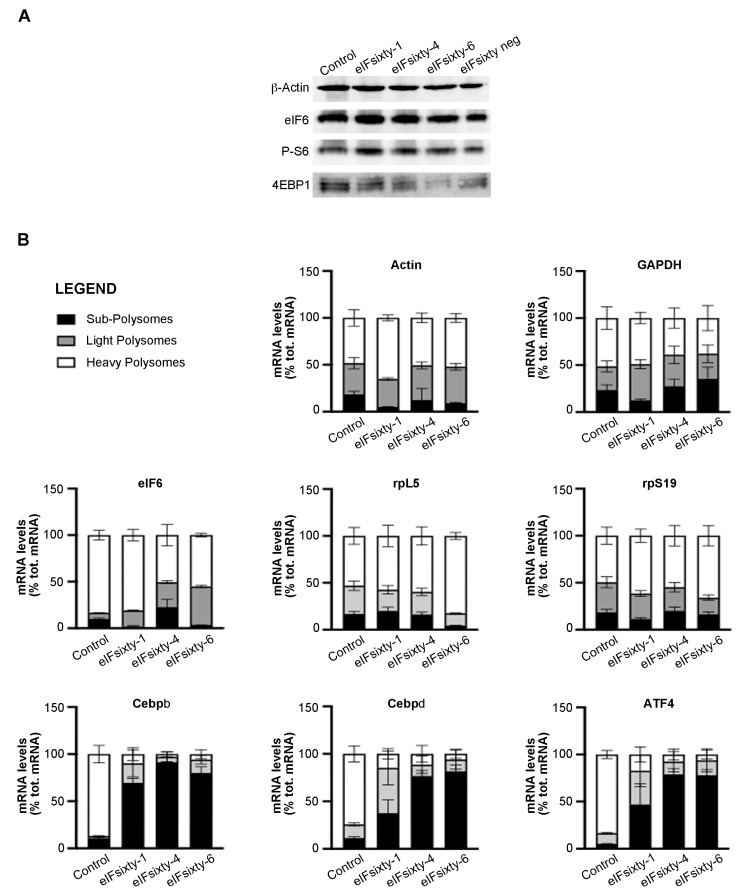
(**A**) Western blots for the activation of downstream targets of mTORC1 in H266 cells after the administration of eIFsixty-i compounds (eIFsixty-1, eIFsixty-4 and eIFsixty-6). Data show that all the compounds do not affect mTORC1 signaling. (**B**) Quantification of mRNA levels in heavy, light, and subpolysomes of H266 cells after the administration of eIFsixty-i compounds (eIFsixty-1, eIFsixty-4, and eIFsixty-6). Data show that all compounds, although to a different extent, induce a strong shift from the polysomal to the subpolysomal peak of eIF6 target mRNAs, i.e., ATF4, CEBPβ, and CEBPδ.

**Figure 8 cells-09-00172-f008:**
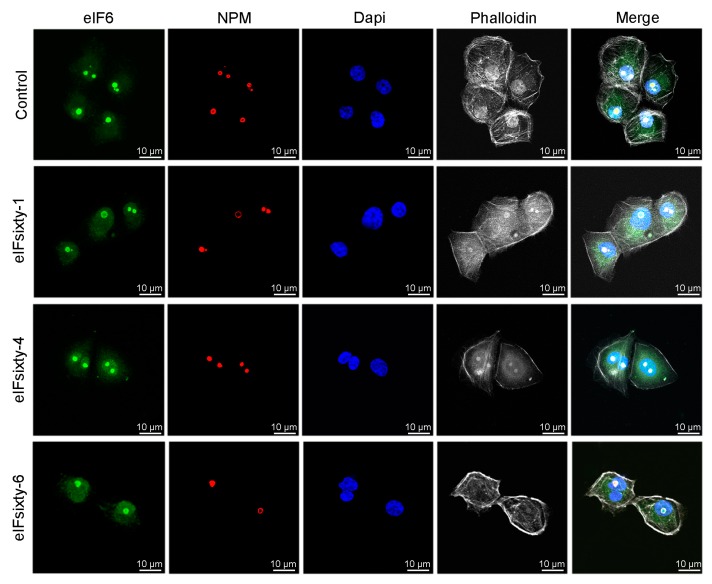
eIFsixty-i compounds do not alter nucleolar integrity. Representative images of eIF6 and NPM-stained nucleoli from H226 cells in the absence (control) or presence of eIFsixty-i compounds (eIFsixty-1, eIFsixty-4, and eIFsixty-6). Cells were treated with eIFsixty-i compounds for 1 h at the IC50 concentration. Actin and nuclei were visualized by incubating fixed cells with phalloidin or DAPI (4′,6-diamidino-2-phenylindole), respectively (scaling bars, 10 µm).

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
