# Peer review of "Discovery and Preliminary Characterization of Translational Modulators that Impair the Binding of eIF6 to 60S Ribosomal Subunits"

_cells, 2020, doi:10.3390/cells9010172_

Round 1
Reviewer 1 Report
Inhibitors of eIF6-ribosome interaction
A large group of authors represented by Stefano Biffo as corresponding author has submitted a manuscript describing a rapid screening assay for identifying chemical inhibitors of interaction between eIF6 and the ribosome. eIF6 has two basic functions as a ribosome biogenesis factor (apparently not relevant to the assay, see Fig 8) and as a translation factor which prevents the 40S and 60S ribosomal subunits from forming inactive 80S subunits without an associated mRNA. Even though eIF6 thus has a general function in translation, it appears to be involved in differentiation of the rate with which different mRNAs associate with the ribosome. As such, eIF6 has also been implicated in the mechanism for various human diseases. It is therefore important to understand the precise mechanism of eIF6 function and to identify drugs of potential clinical manipulation of the interaction.
The current manuscript describes an assay for rapid screening of chemicals that can modify the interaction of eIF6 with the ribosome. As such this is an important manuscript. However, a number of improvements are required in order to give the manuscript its proper impact. In my opinion, that the following issues must be addressed prior to acceptance.
The assay is described in Fig 1A as a FRET assay that brings biotin-tagged ribosomes and His-tagged eIF6 together with beads for primary and secondary light emission. If eIF6 is associated with the ribosome, the energy transfer is active, but if the interaction is inhibited by chemicals, ribosome and eIF6 are not close enough for the energy transfer and the secondary light signals declines. This is a clever assay, but it is not straight forward for a non-aficionado to connect the picture in Fig 1A with the microtiter-plate assay described in Materials and Methods. Where is the microtiter dish in Fig 1A? In contrast the plate is clearly relevant for the ELISA assay in Fig 3. This should be clarified. Line 38: Do the authors mean “differentially decoded”? Lines 44-45 and other places: Please define abbreviations Line 60: “On this line” is awkward. “Along these lines”? Line 142 and other places: I presume that the authors mean nmol, (nano-mole) i.e. amount of the compound, not nM, (nano-molar), i.e. the concentration of the compound. Line 142: It appears that the text should be “biotinylated ribosomes”. Please clearly indicate where tagged and non-tagged compounds are used. Line 215: “confirmed the anticipated behavior”. Refer to specific results. Fig 1D: header should be “non-biotinylated ribosome-eIF6 competition assay” or even “non-biotinylated ribosome-His-tagged eIF6 competition assay” Line 243 and Fig 2A: What is the meaning of a negative value on the X axis? Compounds strengthening the FRET signal? Lines 304-5: What are the statistics for comparing the results for sixty-1 vs -4? The evaluation of the value of the assay depends highly on the understanding of the biochemical mechanism behind the changes in the FRET result. The molecular-biological/biochemical analysis of the three compounds (Figs 4-7) clearly show that the mechanism behind the FRET results is different between the compounds. This may also be the reason for showing the chemicals bound to eFI6 OR to ribosomes (Fig 3, right column of cartoons). There is no discussion of this important aspect. Please provide a discussion as this would be important for interpreting the results and improving the assay.

Author Response
RESPONSE TO REVIEWER 1 COMMENTS
Please find point-to-point responses to reviewer 1 below:
(...) However, a number of improvements are required in order to give the manuscript its proper impact. In my opinion, that the following issues must be addressed prior to acceptance.
We thank you for the comments and the time spent to read the manuscript.
Point 1: The assay is described in Fig 1A as a FRET assay that brings biotin-tagged ribosomes and HistaggedeIF6 together with beads for primary and secondary light emission. If eIF6 is associated with the ribosome, the energy transfer is active, but if the interaction is inhibited by chemicals, ribosome and eIF6 are not close enough for the energy transfer and the secondary light signals declines. This is a clever assay, but it is not straight forward for a non-aficionado to connect the picture in Fig 1A with the microtiter-plate assay described in Materials and Methods. Where is the microtiter dish in Fig 1A? In contrast the plate is clearly relevant for the ELISA assay in Fig 3. This should be clarified.
RESPONSE 1: The binding reaction shown in fig 1A is happening in solution inside each of the wells of the 384 well plate. Microbeads can be dispensed with normal dispensers as they are sufficiently small not to clog the tip of the dispenser. Information has been added in the results paragraph.
Point 2: Line 38: Do the authors mean “differentially decoded”?
RESPONSE 2: Yes. We have corrected the statement.
Point 3: Lines 44-45 and other places: Please define abbreviations
RESPONSE 3: mTORC1 and 4E-BPs have now been defined.
Point 4: Line 60: “On this line” is awkward. “Along these lines”?
RESPONSE 4: We have substituted the phrase accordingly.
Point 5: Line 142 and other places: I presume that the authors mean nmol, (nano-mole) i.e. amount of the compound, not nM, (nano-molar), i.e. the concentration of the compound.
RESPONSE 5: We mean amount, indeed. We have corrected it in the relevant places in the text.
Point 6: Line 142: It appears that the text should be “biotinylated ribosomes”. Please clearly indicate where tagged and non-tagged compounds are used.
RESPONSE 6: The statement has been corrected.
Point 7:Line 215: “confirmed the anticipated behavior”. Refer to specific results.
RESPONSE 7: It refers to the linear relationship between signal and noise. The statement has now been corrected.
Point 8: Fig 1D: header should be “non-biotinylated ribosome-eIF6 competition assay” or even “nonbiotinylated ribosome-His-tagged eIF6 competition assay”
RESPONSE 8: We have wow explained it in the legend.
Point 9: Line 243 and Fig 2A: What is the meaning of a negative value on the X axis? Compounds strengthening the FRET signal?
RESPONSE 9: A negative value refers to compounds which increase the binding of eIF6 to the 60S (which were not pursued).
Point 10: Lines 304-5: What are the statistics for comparing the results for sixty-1 vs -4?
RESPONSE 10: Two sample T-test, equal variance. Statistics have been added in the methods section.
Point 11: The evaluation of the value of the assay depends highly on the understanding of the biochemical mechanism behind the changes in the FRET result. The molecular-biological/biochemical analysis of the three compounds (Figs 4-7) clearly show that the mechanism behind the FRET results is different between the compounds. This may also be the reason for showing the chemicals bound to eFI6 OR to ribosomes (Fig 3, right column of cartoons). There is no discussion of this important aspect. Please provide a discussion as this would be important for interpreting the results and improving the assay.
RESPONSE 11: As the reviewer correctly points out, the compounds may bind either the 60S or eIF6, this is a very important issue and may explain the differences we see. We currently do not have an answer. A small statement has been added in the discussion section, last two paragraphs.
Reviewer 2 Report
The manuscript by Pesce et al. describes a preliminary characterization of a class of compounds that inhibits the binding between eIF6 with the 60S ribosome subunits.
The authors described the effect of these compounds on the modulation of protein synthesis and cell proliferation in different cancer cell lines.
They suggested an intriguing idea that it is possible to modulate the translation of selective mRNAs altering the binding of initiation factor 6 with the large subunit of the ribosome, without affecting the ribosome biogenesis itself.
In my opinion, I found this study interesting and definitely adding significant value to the translation field but it is still quite preliminary as the authors also recognized. The compound screening and biochemical study are clear and well done but the molecular characterization is still weak.
My major concern is about the effect of compound eIFsixty-4 on protein synthesis. eIFsixty-4 shows a very strong translation inhibition either via the puromycin labeling (fig 5) and analyzing the polysome profile (fig 6) suggesting a general translation shut-off. The authors claim that the effect on translation is specific to a class of mRNAs which are well-known targets of eIF6. Moreover, it is not clear how they can explain this discrepancy, how the rest of cellular mRNAs are translated if the translation seems to be dramatically affected?
Nevertheless, I think this is an interesting paper and here some comments and suggestion on how to improve the manuscript:
Can the authors clarify if the puromycin labeling, showed in figure 5A, it is run in one gel or in two separate gels? Since they compared the signal between the different eIFsixty-i and the control, it is essential that the control is loaded on the same gel. Moreover is the control done it in presence of DMSO? How the signal in figure 5B is quantified? The authors showed very convincing data in figure 2 and figure 3 about the effect of eIFsixty-i compounds on the inhibition of the binding between the 60S and eIF6. In light of the in vitro data, how these compounds affect the distribution of eIF6 across the polysomes gradient showed in figure 6? Can the authors provide a western blot for eIF6? I have some concerns regarding figure 7, which plays a very important role in this story. It is essential, in order to conclude, that the effect of these inhibitors is specific on eIF6 target mRNAs, to include more controls in this analysis. For instance, more transcripts like RPL5 are required, which are TOP mRNAs, and more housekeeping mRNAs like Actin, GAPDH, Tubulin, etc. It will make the result more convincing. The effect of eIFsixty-i on ribosome biogenesis is not well described in the text, especially in paragraph 3.8. Can the authors elaborate a bit more on this section? The experiment in the paragraph is not discussed, and either in the result. Moreover, there is not a positive control of how NPM distribution would be affected if the ribosome biogenesis is impaired in H226 cells. The authors justify the outcome of this experiment saying that the ribosome biogenesis is not affected by the compounds, but in my opinion to strengthen this point they could also analyze the level of pre-rRNA 47S via qPCR showing that the transcription of rRNA is not altered, or check the expression of p53 protein or mRNA which is well known to be induced once ribosomal stress is activated via defects in ribosome biogenesis. In the discussion the authors speculate that “An interesting example is represented by rpL5 the lack of effect of eIFsixty-1 and eIFsixty-4 on rpL5 translation may be due to the fact that this mRNA is a TOP mRNA, very abundant and under the control of mTOR signaling, whereas, as far as we know, eIF6 activity is totally independent of mTOR signaling “In order to conclude that mTORC1 is not involved, despite the major inhibition of translation caused by the inhibitors, the authors could provide a western blot for the major targets of mTORC1 like S6K1, RPS6 or 4EBP1.
Minor points:
In the introduction the acronym for mTORC1 is spelled in two different ways: line 44 mTORC1 and line 49, 57, 59 is spelled as mTORc1. Can the authors correct to mTORC1? And similarly, eFL1 is spelled eFL1 line 81 and Efl1 line 84 In material and methods, the section named “Analysis of translated mRNAs” it is not clear why the internal control used was called rp18S/rp18s line 187/188 and later 18s rRNA. I believe that the authors used the 18S as control and not the ribosomal protein S18, is it correct? Can the authors clarify in the text? In the section “ Immunofluorescence” there is no clear information about the antibodies used in the assay, company, dilution and moreover the microscope analysis is not described. As mentioned early also the quantification of puromycin labeling is not well described in the material and method. In paragraph 3.7 line 343 the authors mentioned “see discussion”, but this is not appropriate, I suggest either to remove this sentence or to add more explanation about this important result.
Author Response
RESPONSE TO REVIEWER 2 COMMENTS:
Please find point-to-point responses to reviewer 1 below:
Point 1: (…) The molecular characterization is still weak. My major concern is about the effect of compound eIFsixty-4 on protein synthesis. eIFsixty-4 shows a very strong translation inhibition either via the puromycin labeling (fig 5) and analyzing the polysome profile (fig 6) suggesting a general translation shut-off. The authors claim that the effect on translation is specific to a class of mRNAs which are well-known targets of eIF6. Moreover, it is not clear how they can explain this discrepancy, how the rest of cellular mRNAs are translated if the translation seems to be dramatically affected?
RESPONSE 1: We agree that the molecular characterization is still weak, but we think that, at this stage, it is mandatory to publish this study as a proof-of-concept, in order to be able to secure funding and continue in this direction. We apologize for the misunderstanding on eIFsixty-4, as our interpretation of the data coincides with that of reviewer 2.
This said it is evident that all the compounds have a stronger effect on eIF6 known targets, rather than on general targets, the same being true also for eIFsixty-4. There is no discrepancy in the data. First, cells treated with eIFsixty-4 have more than 50% residual translation. Second, the way the RT-qPCR signal is normalized on gradients is such that you do not evaluate whether translation of an mRNA is reduced in absolute amounts, but whether if that mRNA drops more than the 18S rRNA (which itself shifts if initiation of translation is globally reduced). mRNAs whose initiation drops more than the general drop of translation are reduced on polysomes. Conversely mRNA whose initiation is reduced less than the global rate of translation are apparently increased. mRNAs that drop as much as the general drop of translation remain stable. The large changes in the specific targets, compared to the small changes in the other genes are therefore consistent.
Point 2: Can the authors clarify if the puromycin labeling, showed in figure 5A, it is run in one gel or in two separate gels? Since they compared the signal between the different eIFsixty-i and the control, it is essential that the control is loaded on the same gel. Moreover, is the control done it in presence of DMSO? How the signal in figure 5B is quantified?
RESPONSE 2: All the samples corresponding to the treatments with different compounds cannot be run in triplicate in the same gel simply because they are too many, but the experiments are always made in triplicate with the DMSO-treated control run aside randomizing lanes. Signals are quantified with Image J. Experiments have been performed multiple times and a representative image is shown. This is specified in the methods section.
Point 3: The authors showed very convincing data in figure 2 and figure 3 about the effect of eIFsixty-i compounds on the inhibition of the binding between the 60S and eIF6. In light of the in vitro data, how these compounds affect the distribution of eIF6 across the polysomes gradient showed in figure 6? Can the authors provide a western blot for eIF6?
RESPONSE 3: A blot has been added showing the expected shift of eIF6 to the left after treatment with all compounds (Fig. 6). The value of defining quantitative binding of eIF6 to cytoplasmic 60S in vivo in a polysomal gradient is however limited. I explain it: in normal conditions most eIF6 is already free and the 60S-bound eIF6, found in gradients is due to stable binding to immature 60S subunits [1-5]. Inhibitory compounds are not expected to shift much of the cytoplasmic eIF6 in a gradient because at the steady-state it is already largely free. Indeed, in agreement with 20 years of studies done in our lab, a) eIF6 stably bound to 60S blocks translation, but b) rapid on-off binding of eIF6 is required for efficient translation. In short, the data make total sense.
Point 4: I have some concerns regarding figure 7, which plays a very important role in this story. It is essential, in order to conclude, that the effect of these inhibitors is specific on eIF6 target mRNAs, to include more controls in this analysis. For instance, more transcripts like RPL5 are required, which are TOP mRNAs, and more housekeeping mRNAs like Actin, GAPDH, Tubulin, etc. It will make the result more convincing.
RESPONSE 4: We added some controls (new Fig. 7). Future studies will address by ribosome profiling the whole spectrum of affected mRNAs, but this is out of scope now. We think that the extent of the shift of known eIF6 targets, compared to general targets, is undisputable.
Point 5: The effect of eIFsixty-i on ribosome biogenesis is not well described in the text, especially in paragraph 3.8. Can the authors elaborate a bit more on this section? The experiment in the paragraph is not discussed, and either in the result. Moreover, there is not a positive control of how NPM distribution would be affected if the ribosome biogenesis is impaired in H226 cells. The authors justify the outcome of this experiment saying that the ribosome biogenesis is not affected by the compounds, but in my opinion to strengthen this point they could also analyze the level of pre-rRNA 47S via qPCR showing that the transcription of rRNA is not altered, or check the expression of p53 protein or mRNA which is well known to be induced once ribosomal stress is activated via defects in ribosome biogenesis.
RESPONSE 5: This is true. A statement in the last paragraph of the discussion has been added regarding the need to further address ribosome biogenesis in detail. At this stage we simply claim that short term treatment with eIFsixty does not displace eIF6 from nuclear pre-ribosomes, or affects nucleolar structure. We would like to stress that this is the first proof-of-principle study where we identify compounds inhibiting the binding of eIF6 to the 60S. All proposed studies are important and need to be done, but are not the main point here.
Point 6: In the discussion the authors speculate that “An interesting example is represented by rpL5 the lack of effect of eIFsixty-1 and eIFsixty-4 on rpL5 translation may be due to the fact that this mRNA is a TOP mRNA, very abundant and under the control of mTOR signaling, whereas, as far as we know, eIF6 activity is totally independent of mTOR signaling “ In order to conclude that mTORC1 is not involved, despite the major inhibition of translation caused by the inhibitors, the authors could provide a western blot for the major targets of mTORC1 like S6K1, RPS6 or 4EBP1.
RESPONSE 6: Blots have been added (Figure 7). There is no effect on the phosphorylation of rpS6 or 4E-BP1 which are very good proxyes for mTORC1 activation.
Point 7: Minor points:
In the introduction the acronym for mTORC1 is spelled in two different ways: line 44 mTORC1 and line 49, 57, 59 is spelled as mTORc1. Can the authors correct to mTORC1? And similarly, eFL1 is spelled eFL1 line 81 and Efl1 line 84 In material and methods, the section named “Analysis of translated mRNAs” it is not clear why the internal control used was called rp18S/rp18s line 187/188 and later 18s rRNA. I believe that the authors used the 18S as control and not the ribosomal protein S18, is it correct? Can the authors clarify in the text? In the section “ Immunofluorescence” there is no clear information about the antibodies used in the assay, company, dilution and moreover the microscope analysis is not described. As mentioned early also the quantification of puromycin labeling is not well described in the material and method. In paragraph 3.7 line 343 the authors mentioned “see discussion”, but this is not appropriate, I suggest either to remove this sentence or to add more explanation about this important result.
RESPONSE 7: Thanks. mTORC1, efl1, 18S rRNA have now been corrected. IFL data was added. Quantification of puromycin was also added. "see discussion" has been removed. Concerning immunofluorescence studies, the concentration of the primary antibodies depends on the secondary antibody technology used and varies and for this reason it has not been added. We added new references and microscope analysis.
REFERENCES:
[1] Sanvito, F.; Piatti, S.; Villa, A.; Bossi, M.; Lucchini, G.; Marchisio, P.C.; Biffo, S. The beta4 integrin interactor p27(BBP/eIF6) is an essential nuclear matrix protein involved in 60S ribosomal subunit assembly. J Cell Biol 1999, 144, 823-837, doi:10.1083/jcb.144.5.823.
[2] Ceci, M.; Gaviraghi, C.; Gorrini, C.; Sala, L.A.; Offenhauser, N.; Marchisio, P.C.; Biffo, S. Release of eIF6 (p27BBP) from the 60S subunit allows 80S ribosome assembly. Nature 2003, 426, 579-584, doi:10.1038/nature02160 nature02160 [pii].
[3] Volta, V.; Ceci, M.; Emery, B.; Bachi, A.; Petfalski, E.; Tollervey, D.; Linder, P.; Marchisio, P.C.; Piatti, S.; Biffo, S. Sen34p depletion blocks tRNA splicing in vivo and delays rRNA processing. Biochem Biophys Res Commun 2005, 337, 89-94, doi:S0006-291X(05)02009-7 [pii] 10.1016/j.bbrc.2005.09.012.
[4] Gandin, V.; Miluzio, A.; Barbieri, A.M.; Beugnet, A.; Kiyokawa, H.; Marchisio, P.C.; Biffo, S. Eukaryotic initiation factor 6 is rate-limiting in translation, growth and transformation. Nature 2008, 455, 684-688, doi:nature07267 [pii] 10.1038/nature07267.
[5] Brina, D.; Grosso, S.; Miluzio, A.; Biffo, S. Translational control by 80S formation and 60S availability: The central role of eIF6, a rate limiting factor in cell cycle progression and tumorigenesis. Cell Cycle 2011, 10, doi:17796 [pii].
[6] Russell, D.W.; Spremulli, L.L. Purification and characterization of a ribosome dissociation factor (eukaryotic initiation factor 6) from wheat germ. J Biol Chem 1979, 254, 8796-8800.
[7] Brina, D.; Miluzio, A.; Ricciardi, S.; Clarke, K.; Davidsen, P.; Viero, G.; Tebaldi, T.; Offenhäuser, N.; Rozmann, J.; Rathkolb, B., et al. eIF6 coordinates insulin sensitivity and lipid metabolism by coupling translation to transcription. Nature Communications 2015, 6, 8261.
[8] Calamita, P.; Miluzio, A.; Russo, A.; Pesce, E.; Ricciardi, S.; Khanim, F.; Cheroni, C.; Alfieri, R.; Mancino, M.; Gorrini, C., et al. SBDS-Deficient Cells Have an Altered Homeostatic Equilibrium due to Translational Inefficiency Which Explains their Reduced Fitness and Provides a Logical Framework for Intervention. PLoS Genet 2017, 13, e1006552, doi:10.1371/journal.pgen.1006552.
Reviewer 3 Report
The authors established an assay based on the AlphaScreen technology to measure interactions between eIF6 (a translational initiation factor) and purified 60s ribosomal subunits. They further used a library comprised of 2,977 bioactive compounds to screen for compounds that modulate eIF6-60S interactions in this assay. They found several compounds that inhibit (or increase) interactions and further focused on three compounds for validation. To do so, they first recapitulated the interactions and determined the IC50 values (micromolar range), and they performed ELISA assays. They tested the three selected compounds in cell-viability assays in different cell lines, measuring strong cytotoxic effects for one of the compounds (eIFsixty-4). Translational inhibition was tested upon application of compounds with puromycin labelling experiments, confirming strong effects on translation for at least one compound (eIFsixty-4). More revealing, polysomal profiling showed changes in the translational profile for established eiF6-dependent mRNAs. Finally, the authors showed that nucleolar structure was not affected by all three compounds.
Overall, this is an interesting and timely study that reveals a first set of compounds that likely affect eIF6-60S interactions. The results have been validated with a series of in vitro interactions assays and ex vivo studies in cells showing effects on translation and cell-viability. Regarding the latter, one point that should be interpreted with more care relates to potential off-target activity of these compounds in cells. While the authors postulate that the compounds may alter translation, which could lead to effects on cell-proliferation/ viability, the reverse is also possible as the compounds could lead to off-targets effects that induce a stress response which further promotes translation repression. Essentially, it is not clear whether the seen effects on translation are a cause or a consequence. While sorting this out may be beyond the scope of this paper, these limitations should be brought to the attention of the reader and interpretations modified accordingly. On this line, in vitro translation assays may further bridge the gap between in vitro and ex vivo data.
Specific comments:
1. Assay development. I was wondering about the specificity of the interactions measured in the AlphaScreen assay. It is not clear how well the assay is controlled for specificity and background as negative controls are often absent. For instance, Fig. 1B misses the addition of biotinylated beads (w/o 60S) or a biotinylated unrelated protein to measure background binding. Competition experiments were done with non-biotinylated eIF6 (target), which is good. Nonetheless, the authors should use an unrelated protein (e.g. BSA) as a control. At least one such control experiment should be included.
2. Hit identification. The authors should provide the raw data with activity measurements in the Supplement. Interestingly, as seen in Fig. 2A, there are also compounds that seem to negatively inhibit (-inhibition?) and therefore strengthen interactions between eIF6 and 60S. The authors should mention the result and discuss it.
3. Viability assays. Line 290. The authors should modify/ soften the statement “these data suggest that eIFsixty-4 may be a strong inhibitor of translation”. The compound may also have other effects on cells that could lead to the observed cytotoxicity (see comments above).
4. The same concerns polysomal assays/ puromycin treatment. Since eIFsixty-4 strongly inhibits cell-proliferation, one could expect that translational activity is decreased along with the measured decrease of puromycin incorporation and accumulation of 80S monosomes. The authors should explain that cell-proliferation is directly connected to translational activity. In other words, any compound that inhibits cell-grow leads to reduced translation, irrespective of whether it targets the translational machinery or has other effects i (e.g. on signal transduction pathways that link to translation).
5. The shift in polysomal profiles (Fig. 6) could be quantified by comparing the ratio of 80S to polysome peak areas. It would also be good to display the distribution of rRNAs in the fractions by RT-PCR.
6. Analysis of translated mRNAs (Methods & Results). The analysis is not sufficiently described in the Methods section. Real-time PCR measurements should be described according to guidelines for reporting RT-qPCR data (e.g. #PCR cycles, temp, primer concentration and controls).
7. How was mRNA quantification across the 3 different pools performed by taking 18S rRNA as a standard (Fig. 7)? rRNA is not equally distributed in sucrose gradients (accumulated in monosome/polysomes). It may be better to use equal amounts of a reference RNA spiked into each fraction prior to RNA isolation and use this for normalisation. Furthermore, it should be explained which fractions of the polysomal profile (Fig. 6) were combined to make each pool (either explain in text or graphics in Fig. 6). Again, the authors assume that 18 rRNA is equally present across the 3 pools and does not change after application of the compounds (as used for standardisation and calculations). I am not sure whether this is the case and the authors should add evidence or explain.
Author Response
RESPONSE TO REVIEWER 3 COMMENTS:
Please find point-to-point responses to reviewer 3 below:
(…) Essentially, it is not clear whether the seen effects on translation are a cause or a consequence. While sorting this out may be beyond the scope of this paper, these limitations should be brought to the attention of the reader and interpretations modified accordingly. On this line, in vitro translation assays may further bridge the gap between in vitro and ex vivo data.
Thanks for the comments. Off-target effects are obviously possible. A caution statement in the discussion has been added. Experiments on translation were done with 1-hour treatment, a relatively short time. In vitro translation with eIF6 is very complicated by the fact that, in vitro, the antiassociation activity predominates and is inhibitory [2,6], most likely because we do not completely understand its in vivo dynamic regulation.
Specific comments
Point 1: Assay development. I was wondering about the specificity of the interactions measured in the AlphaScreen assay. It is not clear how well the assay is controlled for specificity and background as negative controls are often absent. For instance, Fig. 1B misses the addition of biotinylated beads (w/o 60S) or a biotinylated unrelated protein to measure background binding. Competition experiments were done with non-biotinylated eIF6 (target), which is good. Nonetheless, the authors should use an unrelated protein (e.g. BSA) as a control. At least one such control experiment should be included.
RESPONSE 1: The reviewer’s comment is well taken. We actually already had this done but did not report it in Fig. 1B. We now added the background level as a horizontal line in the plot. Further, in Fig 1C the y axis is reported as signal-background (i.e. no eIF6) so, technically only the specific signal was blotted. In general, while we value the author’s suggestion we believe that, being impossible to bring the background to zero, what is important for the assay is the demonstration of its specificity, which we show via the competition assay in Fig. 1D.
Point 2: Hit identification. The authors should provide the raw data with activity measurements in the Supplement. Interestingly, as seen in Fig. 2A, there are also compounds that seem to negatively inhibit (-inhibition?) and therefore strengthen interactions between eIF6 and 60S. The authors should mention the result and discuss it.
RESPONSE 2: The point the reviewer is making is intriguing though we believe that most compounds showing a negative regulation are within the normal Gaussian distribution, suggesting that these negative values are simply due to variability. We will add this comment in the discussion. Beside, in order to define increasers of binding, we think that a novel screening should be devised.
Point 3: Viability assays. Line 290. The authors should modify/ soften the statement “these data suggest that eIFsixty-4 may be a strong inhibitor of translation”. The compound may also have other effects on cells that could lead to the observed cytotoxicity (see comments above).
RESPONSE 3: The statement has been modified.
Point 4: The same concerns polysomal assays/ puromycin treatment. Since eIFsixty-4 strongly inhibits cell-proliferation, one could expect that translational activity is decreased along with the measured decrease of puromycin incorporation and accumulation of 80S monosomes. The authors should explain that cell-proliferation is directly connected to translational activity. In other words, any compound that inhibits cell-grow leads to reduced translation, irrespective of whether it targets the translational machinery or has other effects i (e.g. on signal transduction pathways that link to translation).
RESPONSE 4: The reviewer is right. However, the polysome assay and the puromycin assay have been run after treatments of one hour, limiting the indirect effects due to cell proliferation inhibition. This said, what the reviewer says remains true. A statement in the discussion section has been added.
Point 5: The shift in polysomal profiles (Fig. 6) could be quantified by comparing the ratio of 80S to polysome peak areas. It would also be good to display the distribution of rRNAs in the fractions by RT-PCR.
RESPONSE 5: We now have performed quantitation. The experiment was run in triplicate and SD was null. Concerning the rRNA analysis given the low amount of material due to limitations in the amount of the compounds, and the fact that we preferred to run multiple parallel experiments, we had not enough material. The left one was used for blotting eIF6 which gives a good signal because our antibodies are outstanding and the signal is concentrated in few lanes.
Point 6: Analysis of translated mRNAs (Methods & Results). The analysis is not sufficiently described in the Methods section. Real-time PCR measurements should be described according to guidelines for reporting RT-qPCR data (e.g. #PCR cycles, temp, primer concentration and controls).
RESPONSE 6: The requested details have now been added.
Point 7: How was mRNA quantification across the 3 different pools performed by taking 18S rRNA as a standard (Fig. 7)? rRNA is not equally distributed in sucrose gradients (accumulated in monosome/polysomes). It may be better to use equal amounts of a reference RNA spiked into each fraction prior to RNA isolation and use this for normalisation. Furthermore, it should be explained which fractions of the polysomal profile (Fig. 6) were combined to make each pool (either explain in text or graphics in Fig. 6). Again, the authors assume that 18 rRNA is equally present across the 3 pools and does not change after application of the compounds (as used for standardisation and calculations). I am not sure whether this is the case and the authors should add evidence or explain.
RESPONSE 7: How to normalize polysomal fractions is clearly crucial [7,8]. This said, we do not assume that rRNA is equally present, rather this is part of the analysis. We do not add spike-ins because the problem is not the normalization of the extraction, the problem is to understand whether initiation of a given mRNA is reduced more than the global rate of translation. The logic of this experiments is exactly to quantify an mRNA compared to the shifts of the 18S, which allows to detect mRNAs that shift more or less than the 18S itself. mRNAs whose initiation is reduced as much as the 18S (global translation) remain relatively stable. mRNAs that are reduced specifically at initiation are reduced more than the 18S [7].
Each fraction is expressed as a percentage of the total signal. Anyway, by comparing data of actin to the specific targets, it is evident that our procedure works. Finally, these experiments are bound to provide relative shifts of several targets and not absolute amounts.
The pooled fractions are now indicated.
REFERENCES:
[1] Sanvito, F.; Piatti, S.; Villa, A.; Bossi, M.; Lucchini, G.; Marchisio, P.C.; Biffo, S. The beta4 integrin interactor p27(BBP/eIF6) is an essential nuclear matrix protein involved in 60S ribosomal subunit assembly. J Cell Biol 1999, 144, 823-837, doi:10.1083/jcb.144.5.823.
[2] Ceci, M.; Gaviraghi, C.; Gorrini, C.; Sala, L.A.; Offenhauser, N.; Marchisio, P.C.; Biffo, S. Release of eIF6 (p27BBP) from the 60S subunit allows 80S ribosome assembly. Nature 2003, 426, 579-584, doi:10.1038/nature02160 nature02160 [pii].
[3] Volta, V.; Ceci, M.; Emery, B.; Bachi, A.; Petfalski, E.; Tollervey, D.; Linder, P.; Marchisio, P.C.; Piatti, S.; Biffo, S. Sen34p depletion blocks tRNA splicing in vivo and delays rRNA processing. Biochem Biophys Res Commun 2005, 337, 89-94, doi:S0006-291X(05)02009-7 [pii] 10.1016/j.bbrc.2005.09.012.
[4] Gandin, V.; Miluzio, A.; Barbieri, A.M.; Beugnet, A.; Kiyokawa, H.; Marchisio, P.C.; Biffo, S. Eukaryotic initiation factor 6 is rate-limiting in translation, growth and transformation. Nature 2008, 455, 684-688, doi:nature07267 [pii] 10.1038/nature07267.
[5] Brina, D.; Grosso, S.; Miluzio, A.; Biffo, S. Translational control by 80S formation and 60S availability: The central role of eIF6, a rate limiting factor in cell cycle progression and tumorigenesis. Cell Cycle 2011, 10, doi:17796 [pii].
[6] Russell, D.W.; Spremulli, L.L. Purification and characterization of a ribosome dissociation factor (eukaryotic initiation factor 6) from wheat germ. J Biol Chem 1979, 254, 8796-8800.
[7] Brina, D.; Miluzio, A.; Ricciardi, S.; Clarke, K.; Davidsen, P.; Viero, G.; Tebaldi, T.; Offenhäuser, N.; Rozmann, J.; Rathkolb, B., et al. eIF6 coordinates insulin sensitivity and lipid metabolism by coupling translation to transcription. Nature Communications 2015, 6, 8261.
[8] Calamita, P.; Miluzio, A.; Russo, A.; Pesce, E.; Ricciardi, S.; Khanim, F.; Cheroni, C.; Alfieri, R.; Mancino, M.; Gorrini, C., et al. SBDS-Deficient Cells Have an Altered Homeostatic Equilibrium due to Translational Inefficiency Which Explains their Reduced Fitness and Provides a Logical Framework for Intervention. PLoS Genet 2017, 13, e1006552, doi:10.1371/journal.pgen.1006552.
Reviewer 4 Report
In this manuscript, Pesce et al. describe a novel method to screen for translational modulators that impair the binding of eIF6 to 60S ribosomal subunits.
To this end, the authors have optimized eIF6-60S binding assays. Interference screening allowed them to identify three compounds from a bioactive library that inhibit the binding of eIF6 to 60S named “eIFsixty-i”.
The first part of the manuscript is dedicated to the identification but also to the validation of the compounds using an ELISA in vitro Ribosomes Interaction Assay (iRIA) previously developed by the authors. The data presented is relatively clear.
As eIF6 is an anti-association factor that prevents formation of inactive 80S subunits, authors have performed preliminary experiments to evaluate the impact of the eIFsixty-i compounds on translation and ribosome biogenesis. Even though the methods used to this end are not always fully described, this study provides a proof of concept for this screening method and the identification of translational modulators.
Authors should therefore consider correcting a number of imprecisions and improve the following:
The polysomal profiling experiment is not convincing. Polysomes are not clearly visible in the control experiment making it difficult to evaluate the reduction in the polysomal area for the eIFsixty-4. Quantification of 80S:polysome ratios could be added to the figure.
A western blot analysis showing the position of eIF6 in the polysome gradient and the impact of eIFsixty-i compounds on eIF6 localization would be informative.
The polysome profiles of Figure 6 should be annotated to indicate the position of 40S, 60S, 80S and polysomes. The polysome fraction numbers corresponding to heavy, light and subpolysome fractions used for qRT-PCR analysis Figure 7 should be defined explicitly.
The paragraph describing the effects of eIFsixty-i compounds on ribosome biogenesis and nuclear structure is only 3 lines. The rational for this experiment is absent so is the interpretation of the data. Why authors used NPM-staining and detection of Phalloidin in Figure 8 is not mentioned in the text. This has to be corrected. It is otherwise difficult for the reader to understand that ribosomal stress can be induced by impairment of ribosome biogenesis and that the redistribution of nuclear components, such as nucleophosmin, can be considered as a nucleolar stress indicator.
Comparing Material and Methods (line 149) with Figure 3A it is unclear what HRP-conjugated streptavidine was used for.
Line 170: The nature and reference of the antibody used to detect the puromycin labelled proteins should be indicated.
Legend of Figure 5B: Explain how the quantification of the impact of eIFsixty compounds on global translation was performed, the ** p-values and the number of replicates should be clearly indicated.
Line 416: none of the data presented allow concluding for a “lack of effect” on ribosome biogenesis.
Author Response
RESPONSE TO REVIEWER 4 COMMENTS:
Please find point-to-point responses to reviewer 4 below:
In this manuscript, Pesce et al. describe a novel method to screen for translational modulators that impair the binding of eIF6 to 60S ribosomal subunits.To this end, the authors have optimized eIF6-60S binding assays. Interference screening allowed them to identify three compounds from a bioactive library that inhibit the binding of eIF6 to 60S named “eIFsixty-i”.The first part of the manuscript is dedicated to the identification but also to the validation of the compounds using an ELISA in vitro Ribosomes Interaction Assay (iRIA) previously developed by the authors. The data presented is relatively clear.As eIF6 is an anti-association factor that prevents formation of inactive 80S subunits, authors have performed preliminary experiments to evaluate the impact of the eIFsixty-i compounds on translation and ribosome biogenesis. Even though the methods used to this end are not always fully described, this study provides a proof of concept for this screening method and the identification of translational modulators.
Thanks. We appreciate that you see the positive side of this work.
Point 1: Authors should therefore consider correcting a number of imprecisions and improve the following:
The polysomal profiling experiment is not convincing. Polysomes are not clearly visible in the control experiment making it difficult to evaluate the reduction in the polysomal area for the eIFsixty-4. Quantification of 80S:polysome ratios could be added to the figure.
RESPONSE 1: Different cell lines have different polysomal resolutions, which explains the data, together with the fact that cells are analyzed in normal, cycling conditions without particular stimulations. In this condition, the employed cell line has consistently this level of polysomes.
80S/polysome ratios have now been added. As added for reviewer 3, experiments were multiple, SD was minimal, and the amount of the compounds impaired some extra analysis that we normally do (for instance upscale the gradients, test multiple conditions). The effect of eIFsixty-4 is in reality very evident from the incredible increase in 80S.
Point 2: A western blot analysis showing the position of eIF6 in the polysome gradient and the impact of eIFsixty-i compounds on eIF6 localization would be informative.
RESPONSE 2: We added the suggested western blot. All compounds shift eIF6 to the soluble fraction. Please note that a) eIF6 is always absent from polysomes [2] b) part of eIF6 remains bound to nucleolar pre-60S[1]. As explained for another reviewer, the analysis of eIF6 distribution, in vivo, is less informative than what one normally thinks. First, in vivo, most eIF6 is always free. Second, in vitro administration totally blocks 80S. In vivo, eIF6 levels are critical to active initiation, through rapid on-off binding. And last but not least, what happens in vivo on mature ribosomes is not so clear. In vitro, if you add eIF6 to mature 60S it binds at high affinity and it is stable for days. In vivo, if you overexpress eIF6, you see stable binding in around 15% of extra-ribosomes, only in starving conditions. If you stimulate translation, extra-eIF6 supports increased translation, but you do not see much more of it in the 60S peak. In short, dynamic in vivo binding is different from in vitro binding. For this same reason, our results are tremendously exciting.
Point 3: The polysome profiles of Figure 6 should be annotated to indicate the position of 40S, 60S, 80S and polysomes. The polysome fraction numbers corresponding to heavy, light and subpolysome fractions used for qRT-PCR analysis Figure 7 should be defined explicitly.
RESPONSE 3: Annotation was added.
Point 4: The paragraph describing the effects of eIFsixty-i compounds on ribosome biogenesis and nuclear structure is only 3 lines. The rational for this experiment is absent so is the interpretation of the data. Why authors used NPM-staining and detection of Phalloidin in Figure 8 is not mentioned in the text. This has to be corrected. It is otherwise difficult for the reader to understand that ribosomal stress can be induced by impairment of ribosome biogenesis and that the redistribution of nuclear components, such as nucleophosmin, can be considered as a nucleolar stress indicator.
RESPONSE 4 : We agree with the comment. Taken together with other comments on this point, we now have added a statement in the discussion that relates to the need of further studies, which is the real issue. This said, clearly these compounds do not disrupt the binding to nucleolar preribosomes.
Point 5: Comparing Material and Methods (line 149) with Figure 3A it is unclear what HRP-conjugated streptavidine was used for.
RESPONSE 5: We apologize. Our assay works equally well by either labelling eIF6 by biotin, or 60S by biotin, or tagging eIF6 in the N-terminus and detecting it with an antibody. All these things are done in the lab, depending on the detection that we use. The affinities, on-off rates are actually almost identical. In the validation phase, we used all methods to be sure. At this stage it becomes irrelevant. We removed Fig. 3A.
Point 6: Line 170: The nature and reference of the antibody used to detect the puromycin labelled proteins should be indicated.
RESPONSE 6: We now have added this information.
Point 7: Legend of Figure 5B: Explain how the quantification of the impact of eIFsixty compounds on global translation was performed, the ** p-values and the number of replicates should be clearly indicated.
RESPONSE 7: Each experiment was run three or four times (depending from the compound), each time in triplicate, and in reality also at different concentrations in different days (data not shown). Results were quite consistent. We have shown data from one of these experiments run in triplicate. The experiment is straightforward, a) add the compound, b) add puromycin, c) lysis, d) blot.
Point 8: Line 416: none of the data presented allow concluding for a “lack of effect” on ribosome biogenesis.
RESPONSE 8: This is true. We now have corrected the statement, indicating the need for further studies.
REFERENCES:
[1] Sanvito, F.; Piatti, S.; Villa, A.; Bossi, M.; Lucchini, G.; Marchisio, P.C.; Biffo, S. The beta4 integrin interactor p27(BBP/eIF6) is an essential nuclear matrix protein involved in 60S ribosomal subunit assembly. J Cell Biol 1999, 144, 823-837, doi:10.1083/jcb.144.5.823.
[2] Ceci, M.; Gaviraghi, C.; Gorrini, C.; Sala, L.A.; Offenhauser, N.; Marchisio, P.C.; Biffo, S. Release of eIF6 (p27BBP) from the 60S subunit allows 80S ribosome assembly. Nature 2003, 426, 579-584, doi:10.1038/nature02160 nature02160 [pii].
[3] Volta, V.; Ceci, M.; Emery, B.; Bachi, A.; Petfalski, E.; Tollervey, D.; Linder, P.; Marchisio, P.C.; Piatti, S.; Biffo, S. Sen34p depletion blocks tRNA splicing in vivo and delays rRNA processing. Biochem Biophys Res Commun 2005, 337, 89-94, doi:S0006-291X(05)02009-7 [pii] 10.1016/j.bbrc.2005.09.012.
[4] Gandin, V.; Miluzio, A.; Barbieri, A.M.; Beugnet, A.; Kiyokawa, H.; Marchisio, P.C.; Biffo, S. Eukaryotic initiation factor 6 is rate-limiting in translation, growth and transformation. Nature 2008, 455, 684-688, doi:nature07267 [pii] 10.1038/nature07267.
[5] Brina, D.; Grosso, S.; Miluzio, A.; Biffo, S. Translational control by 80S formation and 60S availability: The central role of eIF6, a rate limiting factor in cell cycle progression and tumorigenesis. Cell Cycle 2011, 10, doi:17796 [pii].
[6] Russell, D.W.; Spremulli, L.L. Purification and characterization of a ribosome dissociation factor (eukaryotic initiation factor 6) from wheat germ. J Biol Chem 1979, 254, 8796-8800.
[7] Brina, D.; Miluzio, A.; Ricciardi, S.; Clarke, K.; Davidsen, P.; Viero, G.; Tebaldi, T.; Offenhäuser, N.; Rozmann, J.; Rathkolb, B., et al. eIF6 coordinates insulin sensitivity and lipid metabolism by coupling translation to transcription. Nature Communications 2015, 6, 8261.
[8] Calamita, P.; Miluzio, A.; Russo, A.; Pesce, E.; Ricciardi, S.; Khanim, F.; Cheroni, C.; Alfieri, R.; Mancino, M.; Gorrini, C., et al. SBDS-Deficient Cells Have an Altered Homeostatic Equilibrium due to Translational Inefficiency Which Explains their Reduced Fitness and Provides a Logical Framework for Intervention. PLoS Genet 2017, 13, e1006552, doi:10.1371/journal.pgen.1006552.
Round 2
Reviewer 3 Report
The authors have addressed most of the comments adequately. Nevertheless, I am still confused about the activity profile shown in Fig. 2A that shows data occurrences on both tails (+/- inhibition) of the Gaussian distribution. This may come from variability as the authors suggest. The authors should make the data available as a Supplement so the reader can look this up.
Author Response
Unfortunately, due to IP policies it is not possible to disclose the structure of compounds that are not directly related to the goal of the project that is to identify and characterize compounds that displace the binding of 60S and eIF6.